# Fate of dissolved black carbon in the deep Pacific Ocean

Youhei Yamashita [1,2✉], Motohiro Nakane[2], Yutaro Mori[2], Jun Nishioka [2,3] & Hiroshi Ogawa [4]

Black carbon (BC), a byproduct of biomass and fossil fuel combustion, may impact the climate because it can be stored on Earth's surface for centuries to millennia. Dissolved BC (DBC) occurs ubiquitously in the ocean. However, the DBC cycle in the ocean has not been well constrained. Here, we show the basin-scale distribution of DBC in the Pacific Ocean and find that the DBC concentrations in the deep Pacific Ocean decrease along with deep-ocean meridional circulation. The DBC concentration is negatively correlated with apparent oxygen utilization, a proxy of the integrated flux of sinking particles, in the deep Pacific Ocean, implying that DBC is removed from the deep ocean to abyssal sediments through sorption onto sinking particles. The burial flux of BC to abyssal sediments is estimated to be 0.040–0.085 PgC yr$^{-1}$, corresponding to 1.5–3.3% of the anthropogenic $CO_2$ uptake by the ocean.

---

[1] Faculty of Environmental and Earth Science, Hokkaido University, Sapporo, Japan. [2] Graduate School of Environmental Science, Hokkaido University, Sapporo, Japan. [3] Pan-Okhotsk Research Center, Institute of Low Temperature Science, Hokkaido University, Sapporo, Japan. [4] Atmosphere and Ocean Research Institute, The University of Tokyo, Kashiwa, Japan. ✉email: yamashiy@ees.hokudai.ac.jp

Pyrogenic carbon (PyC) is a series of pyrolyzed products derived from the incomplete combustion of biomass and fossil fuels, and a portion of PyC can be stored for centuries to millennia on the Earth's surface due to its refractory structure[1–5]. The annual global production of PyC as char, charcoal and ash from landscape fires to onsite soils, which are the major global sources of PyC[6], is 0.26 PgC[7]. The annual emission of PyC as soot in smoke to the atmosphere is one to two orders of magnitude smaller than that to onsite soils[8]. These production fluxes of PyC correspond to ~10% of the land sink of anthropogenic $CO_2$[9], implying that PyC can be a significant sink for atmospheric $CO_2$ and can impact the global carbon cycle as well as the climate[2–7]. However, the PyC flux is routinely overlooked in studies on the global carbon budget[7] because the fate of PyC is less constrained than other components of the budget.

The chemical structure and biogeochemical reactivity of PyC are highly heterogeneous and depend on the charring temperature[4,10–13]. A portion of the PyC stored in soils (char, charcoal and ash) is degraded, even though the global flux has not been well constrained[7]. Char and charcoal produce a substantial amount of leachate that is microbially labile[14,15], and some microbially labile PyC, namely, dehydrated carbohydrates (anhydrosugars), has been observed in rivers[16].

The less reactive, condensed aromatic fractions of PyC are defined as black carbon (BC). BC can be determined via the benzenepolycarboxylic acid (BPCA) method, which measures BPCAs, the oxidation products of condensed aromatic fractions[13]. The particulate and dissolved forms of BC (PBC and DBC, respectively) are transferred from soils to rivers and reach the ocean through the land-to-ocean aquatic continuum[17,18]. The fluxes of PBC and DBC from rivers to oceans are estimated to be 0.017–0.037 PgC per year and 0.027 PgC per year, respectively[17,18]. The DBC flux was recently updated to 0.018 PgC per year by Jones et al.[19]. The sum of these riverine fluxes accounts for 13–21% of the annual production of PyC (char, charcoal and ash). The atmospheric deposition of soot is the other

pathway by which BC is input to the ocean (0.012 PgC per year)[20]. Riverine PyC in the particulate fraction, including PBC, is deposited mainly on continental margins[17,21], while riverine DBC is likely photodegraded by sunlight[22,23]. It has recently been suggested that the DBC in the open ocean does not predominantly originate from rivers based on evidence of a large mismatch of the stable carbon isotopic signature between riverine DBC (~−28 to −31‰) and oceanic DBC (~−22 to −25‰)[24]. This evidence suggests that DBC derived from rivers is removed in continental margins before reaching the open ocean. Based upon the isotopic signatures, autochthonous production in the open ocean, hydrothermal origin, contribution of petrogenic organic carbon and input from atmospheric deposition of combustion-derived aerosols were discussed as possible sources of DBC in the open ocean, but the source has not been elucidated[24].

DBC in the deep ocean has been considered to be extremely stable and have cycles of >$10^5$ years because the apparent radiocarbon age of DBC in the deep ocean is calculated to be >20,000 years[25], which is older than that (~6000 years) of bulk dissolved organic carbon (DOC)[26]. On the other hand, Coppola et al.[27] suggested the removal of oceanic DBC through sorption to sinking particles with a flux of 0.016 PgC per year from differences in the apparent radiocarbon age among DBC in ultrafiltered DOC, PBC in sinking particles and abyssal sediments. As such, knowledge of the DBC cycle in the ocean is limited, even though the deep ocean can sequester carbon in the form of DBC along with increased biomass burning and fossil fuel combustion. To evaluate the role of DBC in the deep ocean in the global PyC cycle, the input/output fluxes and the residence time of DBC in the deep ocean should be well constrained. This knowledge may be obtained from the basin-scale distribution along with the meridional overturning circulation in the deep ocean[28,29].

To obtain new insights into the dynamics of DBC in the deep ocean, namely, the input and/or output fluxes, this study determined the full-depth spatial distribution of DBC in the western and central Pacific Ocean, covering the area from 40°S to 54°N (Fig. 1a).

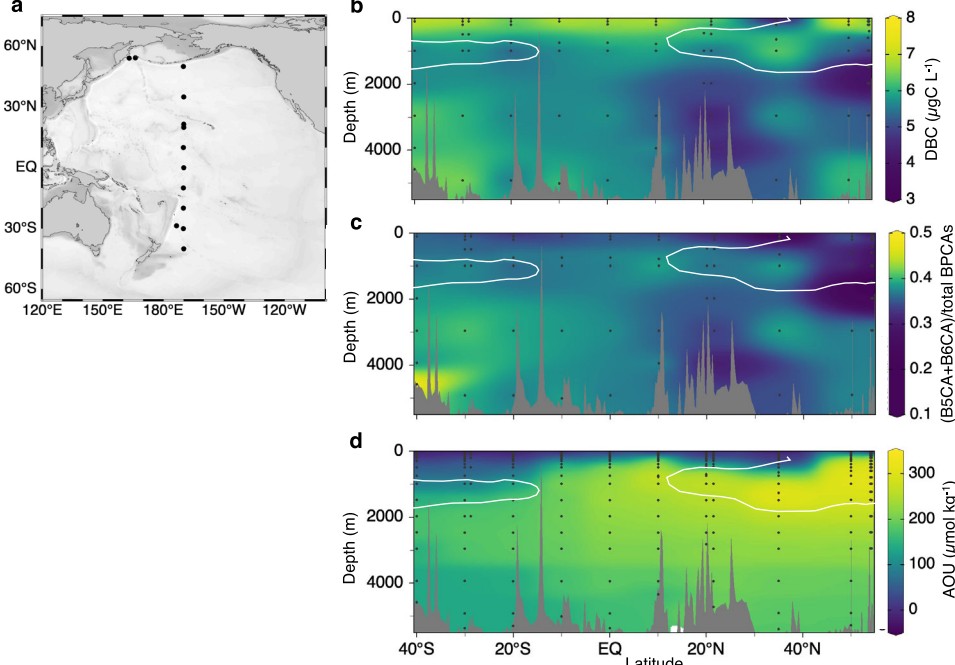

**Fig. 1 Meridional basin-scale distributions of dissolved black carbon (DBC) and apparent oxygen utilisation (AOU) in the Pacific Ocean. a** Station locations. **b** DBC concentration (μgC L$^{-1}$). **c** The ratio of benzenepentacarboxylic acid (B5CA) + benzenehexacarboxylic acid (B6CA) to total benzenepolycarboxylic acids (BPCAs), which is an index of the condensation degree of DBC. **d** AOU (μmol kg$^{-1}$). The white solid lines in **b**, **c** and **d** represent the subsurface salinity minima (salinity = 34.5).

DBC was determined at the molecular level with the BPCA method[18,30–33]. Benzenetricarboxylic acids (B3CA), benzenetetracarboxylic acid (B4CA), benzenepentacarboxylic acid (B5CA) and benzenehexacarboxylic acid (B6CA) were quantified to determine DBC concentrations[30]. The ratio of B5CA and B6CA to total BPCAs ((B5CA + B6CA)/total BPCAs ratio), an index of the condensation degree of DBC[22,32,34,35], was also determined to evaluate the compositional changes in DBC in the deep ocean. A larger ratio is associated with a higher molecular weight, highly condensed DBC and vice versa. The basin-scale distribution of DBC in the deep Pacific Ocean reveals that highly condensed DBC is preferentially removed through sorption onto sinking particles along with the pathway of deep-ocean meridional circulation.

## Results and discussion

**Distribution of DBC in the deep Pacific Ocean.** The DBC concentrations in the surface water of the Pacific Ocean were generally higher than those in the intermediate water (26.6–27.5 $\sigma_\theta$[36–39]; ~200–1500 m, depending on the oceanic region) and deep water (>27.5 $\sigma_\theta$; ~>1500 m) (Fig. 1b and Supplementary Fig. 1a). The vertical pattern of the DBC concentration was similar to those obtained in previous regional observations[24,31,34]. The present observations, however, clarified the meridional variations in the DBC concentration in the Pacific Ocean (Fig. 1b). The DBC concentrations in the intermediate water were relatively low and high compared with those in the deep water in the South and North Pacific Oceans, respectively. Such relatively low and high DBC concentrations were accompanied by low-salinity water, implying that the surface DBC is transported into the mesopelagic layer by Antarctic Intermediate Water (AAIW) and North Pacific Intermediate Water (NPIW), respectively (Supplementary Fig. 1b). The transport of relatively low levels of DBC by AAIW and high levels of DBC by NPIW were consistent with previous observations and have been suggested to result from extensive photodegradation[31] and the transport of DBC from shelf sediments[40], respectively.

In deep water, which consists of Circumpolar Deep Water (CDW) and North Pacific Deep Water (NPDW) (Supplementary Fig. 1a)[36], relatively high DBC levels were evident in the lower part of the subtropical South Pacific Ocean (30°S–40°S) and coincided with relatively low values of apparent oxygen utilisation (AOU), which is the amount of the oxygen consumed by respiration (Fig. 1b, d). The DBC concentration gradually decreased from south to north, accompanied by increases in the AOU, along with the pathway of deep-ocean meridional circulation (Fig. 1 and Supplementary Fig. 1a). A major origin of DBC in CDW is related to the meridional overturning circulation and is therefore associated with North Atlantic Deep Water (NADW) and Antarctic Bottom Water (AABW)[41]. At least some of the DBC in AABW is reported to be derived from Antarctic shelf sediments[42]. The DBC concentrations in the deep water in the subtropical South Pacific (Fig. 1b) are similar to those in CDW in the Southern Ocean (5.9 ± 0.8 µgC L−1), which were determined by the same BPCA method[42]. The decrease in the DBC concentration with deep-ocean meridional circulation (Fig. 1b) is consistent with a recent study that observed higher DBC concentrations in younger NADW than in older, deep water in the North Pacific[24] and is also similar to the observed decrease in DOC concentrations[28,43,44].

The condensation degree of DBC estimated by (B5CA + B6CA)/total BPCAs ratio[32,40] was generally lower in the surface water than in the intermediate and deep waters (Fig. 1c). Such a vertical pattern of the ratio is likely the result of photodegradation because the condensation degree of DBC is known to decrease as

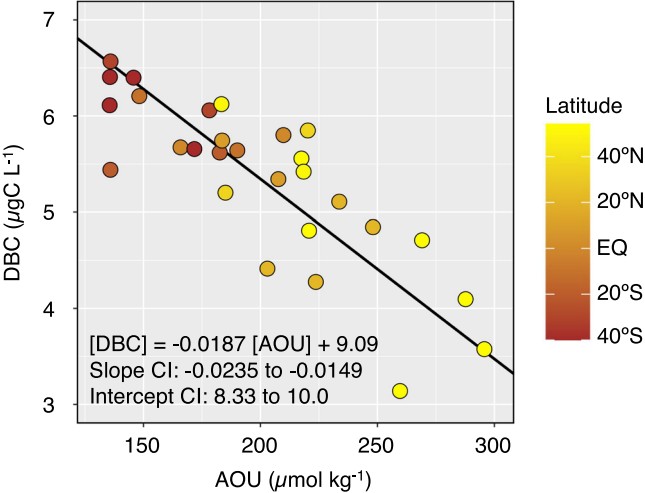

**Fig. 2 Relationship between dissolved black carbon (DBC) concentration and apparent oxygen utilisation (AOU) in deep water (>27.5 $\sigma_\theta$).** A standard major axis regression was applied to determine the linear relationship between the DBC concentration and AOU and is shown as a black line. The 95% confidence interval (CI) determined for the slope and the intercept are also described in the figure. The result of ordinary least squares regression was as follows: [DBC] = −0.0154 [AOU] + 8.42 ($R^2 = 0.676$, $p < 0.001$, $n = 28$). The p value was derived from a two-tailed test.

DBC is photodegraded[22,24]. Interestingly, the (B5CA + B6CA)/total BPCAs ratio tended to decrease from south to north in the deep water.

**DBC sorption onto sinking particles.** Dilution with conservative water mass mixing, microbial degradation and sorption onto sinking particles are possible mechanisms that explain the decrease in DBC observed along with the pathway of deep-ocean meridional circulation. It has been reported that the decrease in DOC concentration in the deep Pacific Ocean along with deep-ocean meridional circulation can be explained by conservative mixing of south-flowing NPIW and north-flowing CDW (also south-flowing NPDW and north-flowing CDW) with some localised sink[44]. Such a dilution of DOC concentration in the deep Pacific Ocean is due to the lower DOC concentration in NPIW compared to CDW. However, DBC concentrations in NPIW are generally higher than those in deep water (Fig. 1b). Thus, dilution with conservative mixing is probably not the main factor behind the measured decrease in the DBC concentration.

In the deep water, the DBC concentration was negatively linearly related to AOU (Fig. 2). The condensation degree of DBC estimated by (B5CA + B6CA)/total BPCAs ratio in the deep ocean decreased from south to north (Fig. 1c). An ordinary least squares regression analysis for the deep water showed that B5CA + B6CA was negatively linearly related to AOU (Fig. 3b and its caption), while B3CA + B4CA was not significantly related to AOU (Fig. 3a and its caption).

For polycyclic aromatic hydrocarbons in surface water, microbial degradation and sorption onto sinking particles have been suggested as the major drivers of the removal of compounds with lower and higher molecular weights, respectively[45,46]. Therefore, if deep water DBC is subjected to microbial degradation, B3CA + B4CA, which is preferentially derived from low-molecular-weight (less condensed) DBC, is expected to decrease with increasing AOU, as a function of oxygen utilisation rate and time after the formation of deep water. However, the lack

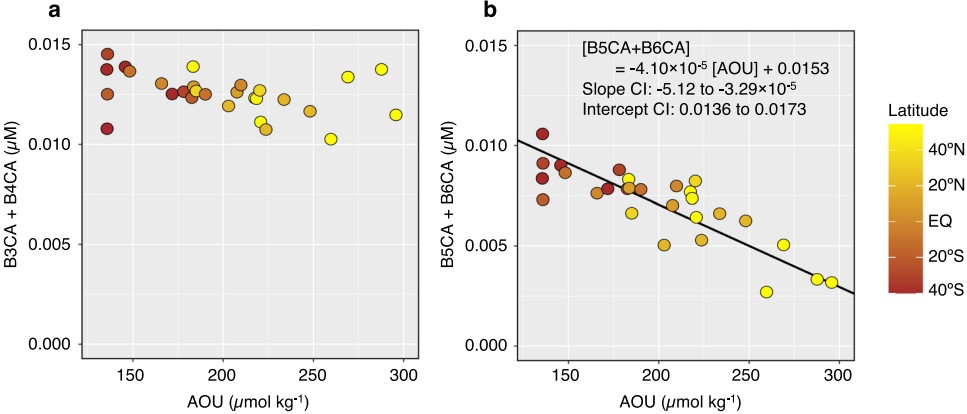

**Fig. 3 Relationships between benzenepolycarboxylic acids (BPCAs) and apparent oxygen utilisation (AOU) in deep water (>27.5 $\sigma_\theta$). a** Relationship between the benzenetricarboxylic acids + benzenetetracarboxylic acid (B3CA + B4CA) concentration and AOU. The result of ordinary least squares regression was not significant ($R^2 = 0.132$, $p = 0.057$, $n = 28$). The $p$ value was derived from a two-tailed test. **b** Relationship between the benzenepentacarboxylic acid + benzenehexacarboxylic acid (B5CA + B6CA) concentration and AOU. Standard major axis regression was applied to determine the linear relationship between the B5CA + B6CA concentration and AOU and is shown as a black line. The 95% confidence interval (CI) determined for the slope and the intercept are also described in the figure. The result of ordinary least squares regression was as follows: [B5CA + B6CA] $= -3.42 \times 10^{-5}$ [AOU] $+ 0.0139$ ($R^2 = 0.695$, $p < 0.001$, $n = 28$). The $p$ value was derived from a two-tailed test.

of a significant relationship between B3CA + B4CA and AOU (Fig. 3a) implies that microbial degradation is probably not a major factor controlling the DBC concentration in the deep ocean. This conclusion is supported by the results of controlled experiments that found that the DBC derived from laboratory-prepared chars had bio-refractory characteristics[15].

Coppola et al.[27] reported that BC in a sinking particle sample collected from the deep ocean is richer in B5CA + B6CA than in ultrafiltered dissolved organic matter, likely due to preferential sorption of the hydrophobic, highly aromatic fraction of DBC to sinking particles. Thus, the negative relationships between B5CA + B6CA and AOU (Fig. 3b) as well as DBC and AOU (Fig. 2) are probably the result of DBC removal through sorption onto sinking particles. AOU represents the total amount of oxygen consumed after the formation of deep water, mainly through the microbial degradation of sinking particulate organic carbon (POC) rather than DOC[47]. AOU is also likely related to the integrated volume of sinking particles that pass through the water mass. Since a substantial fraction of the sinking particles that pass through the water mass degrade before reaching the sediment, it is reasonable to explain the DBC-AOC relationship (Fig. 2) with a two-component model of sinking particles[48,49], i.e., fast-sinking and slow-sinking particles. AOU increases mainly with the degradation of slow-sinking particles, while DBC is mostly removed through sorption onto fast-sinking particles that settle without undergoing extensive degradation. The affinity of the fast-sinking particles for hydrophobic DOC, including DBC, attenuates with increasing water depth due to their sorption, similar to how sinking POC flux decreases with increasing water depth due to their degradation[48], contributing to the apparent relationship between DBC and AOU.

The removal mechanisms of refractory DOC from the deep water have been thus far unknown[44,50,51]. The removal process through sorption onto sinking particles may occur not only for DBC but also for bulk DOC, particularly its hydrophobic fraction, suggesting that sorption of hydrophobic DOC onto sinking particles is possibly a factor affecting the DOC concentration in the deep ocean. However, the evidence provided by regression analyses (Figs. 2 and 3) is not enough to rule out contributions of other processes, namely, mixing and microbial degradation, to the removal mechanisms. Further studies are needed to constrain the removal mechanisms of DBC as well as DOC.

**Implications of DBC fate in PyC cycle.** The mechanism behind the quantitative DBC-AOU relationship (Fig. 2) is based on the proportional relationship between the net sorption rate of DBC onto sinking particles and the increasing rate of AOU and is irrespective of depth differences in the deep water. Thus, analysing the basin-scale relationship between the DBC concentration and AOU with the respiration rate in the open ocean[52] allows us to evaluate the net sorption flux of DBC onto sinking particles or the removal flux of DBC from the abyssal layer (>1000 m) to the sediment (Table 1). The annual removal flux of DBC from the abyssal layer to the sediment was estimated to be 0.0023–0.0046 PgC per year. The removal flux obtained herein was slightly larger than that of the global atmospheric deposition of DBC (i.e., water-soluble BC in aerosols) to the ocean[53] (Fig. 4).

The DBC-AOU relationship in the mesopelagic layer (200–1000 m) is not as simple as that in the abyssal layer, where the influence of water mass mixing is relatively small. The negative relationship may be masked by complex mixing of intermediate waters (AAIW and NPIW) with other water masses (deep water and overlaying water masses), all of which have different levels of surface water-derived DBC. However, removal of DBC through sorption onto sinking particles most likely occurs in not only the abyssal layer but also the mesopelagic layer. Assuming that the same proportional relationship between DBC and AOU is applicable to the mesopelagic layer, the total removal flux from the ocean interior can be calculated as 0.040–0.085 PgC per year (Table 1). This estimate is of the same order of magnitude as the DBC removal flux (0.016 PgC per year) estimated by Coppola et al.[27] with limited data, i.e., PBC/POC % in a sinking particle sample and POC flux to the deep ocean, confirming that the DBC removal through sorption onto sinking particles may be a quantitatively critical sink for DBC in the ocean (Fig. 4). The removal flux was two to three times higher than the total DBC input flux to the ocean[19,53], indicating an obviously inconsistent mass balance (Fig. 4).

The residence times of DBC in the abyssal layer, calculated from the removal flux and the pool size, are 2074–4148 years (Table 1). Although residence time cannot explain the DBC apparent radiocarbon ages of >20,000 years measured in the deep ocean[25], the residence times obtained herein are longer than the time scale of meridional circulation in the deep ocean[54]. The

**Table 1 Global removal flux and residence time of dissolved black carbon (DBC).**

| Layer | Respiration[52] (PgC year$^{-1}$) | Removal of DBC (PgC year$^{-1}$) | DOC pool[43] (PgC) | DBC pool (PgC) | DBC residence time (year) |
|---|---|---|---|---|---|
| Mesopelagic (200–1000 m) | 21–28 | 0.038–0.080 | 138 | 2.76 | 35–73 |
| Abyssal (1000 m–bottom) | 1.3–1.6 | 0.0023–0.0046 | 477 | 9.54 | 2074–4148 |
| Ocean interior (200 m–bottom) | 22–30 | 0.040–0.085 | 615 | 12.3 | 145–308 |

The number range represents the low and high estimates. DOC represents dissolved organic carbon.

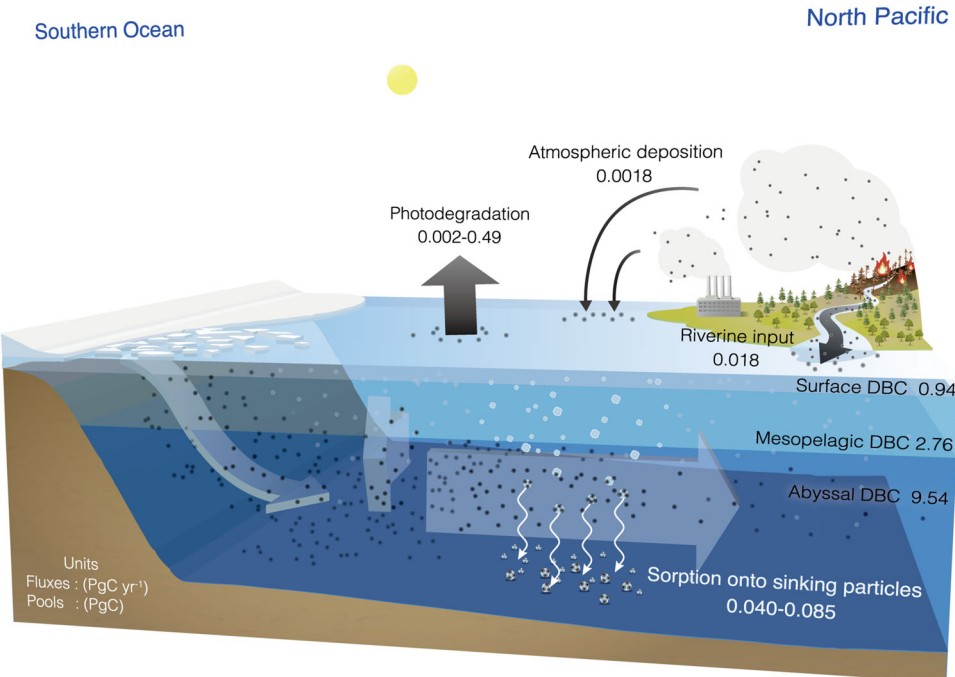

**Fig. 4 Schematics of the dissolved black carbon (DBC) cycle in the ocean, with emphasis on the Pacific Ocean.** The pool sizes of surface DBC, mesopelagic DBC and abyssal DBC were estimated from the dissolved organic carbon (DOC) pool size[43] and the DBC contribution to DOC[31] (Table 1). The riverine input flux was obtained from Jones et al.[19]. The atmospheric deposition flux is the dry deposition flux of water-soluble black carbon (BC) in aerosols[53]. The photodegradation flux was obtained from Stubbins et al.[22]. The flux of sorption onto sinking particles was estimated in this study (Table 1). The units of the pool sizes and fluxes are PgC and PgC year$^{-1}$, respectively.

relatively long residence time and limited removal process of DBC in the abyssal layer indicate that the size of the abyssal DBC pool can increase depending on the strength of the source. To assess the changes in the size of the abyssal DBC pool that occur with changes in combustion activities, measurements of the input flux are necessary. The radiocarbon and stable carbon isotopic signatures of oceanic DBC[24,25] together with the inconsistent mass balance of DBC (Fig. 4) suggest that the source(s) of DBC in which radiocarbon and stable carbon are depleted and enriched compared to those of riverine DBC, respectively, is probably important as the input flux to maintain the oceanic DBC pool.

Since the major fraction of riverine PBC is deposited on continental margins[17], the deposition of BC through the DBC pool is a quantitatively important pathway by which BC enters abyssal sediment. The burial flux of BC to abyssal sediment (Fig. 4) is equivalent to 15–33% of the global PyC (char, charcoal and ash) production resulting from landscape fires[7] and 1.5–3.3% of the anthropogenic $CO_2$ uptake by the ocean[9]. Furthermore, the burial flux including the continental margins is equivalent to up to 47% of the global PyC production[7] and 4.7% of the anthropogenic $CO_2$ uptake by the ocean[9], indicating that the ocean floor can act as a quantitatively important BC storage in the global PyC cycle as well as the global carbon budget.

In summary, our results of basin-scale observations of DBC in the Pacific Ocean provide evidence of DBC removal from the deep ocean through sorption onto sinking particles. The global removal flux from the ocean inferior to abyssal sediments is estimated to be larger than the known global DBC input flux via rivers and atmospheric deposition. Therefore, further studies are required to identify the sources of DBC in the ocean in order to better understand the global PyC cycle.

## Methods

**Samplings and solid-phase extraction**. Observations in the equator and in the South Pacific were carried out in December 2013–January 2014 as part of the R/V *Hakuho Maru* cruise (KH-13-7). Observations in the North Pacific along 170°W were conducted in July 2014 by R/V *Hakuho Maru* (KH-14-3), and observations at a station located at 21.5°N, 170°W were carried out in September 2017 by R/V *Hakuho Maru* (KH-17-4). Observations were carried out at two stations in the western subarctic gyre in August 2018 by the R/V *Professor Multanovskiy* (Mu18). Salinity and temperature were measured using a conductivity-temperature-depth

(CTD) sensor. Dissolved oxygen (DO) concentrations were determined using an oxygen sensor connected to a CTD; this sensor was calibrated using DO concentrations determined by the Winkler titration method. Oxygen solubility was calculated using the Weiss[55] function considering salinity and potential temperature. The apparent oxygen utilisation (AOU) was calculated as the difference between the saturated and measured DO concentrations. The temperature, salinity, DO and dissolved black carbon (DBC) results from the R/V *Professor Multanovskiy* (Mu18) survey can be found in another published work[40].

Seawater samples were collected from the surface layer to the deep layer with 12-L acid-cleaned, Teflon-coated Niskin-X bottles that were mounted on the CTD sensor with a carousel multisampling system. Approximately 9 L of seawater was filtered using a pre-combusted (450 °C, 3–5 h) Whatman GF/F filter (with a 0.7-μm nominal pore size). Immediately after sampling, the pH of the filtrate was adjusted to 2 with HCl. The acidified filtrate was then subjected to solid-phase extraction (SPE) with an SPE cartridge (1 g, Bond Elut PPL, Agilent Technologies), and the SPE cartridge was stored frozen (at −20 °C) in the dark according to the methods described in Dittmar et al.[56] and Nakane et al.[32].

**Analytical procedure for dissolved black carbon**. On land, in a laboratory, the organic matter adsorbed onto and concentrated by the SPE was eluted using methanol, according to the method described in Dittmar et al.[56]. The eluate was adjusted to 10 ml, poured into a glass vial with a Teflon-lined cap, and then stored in a dark in a freezer (at −20 °C).

Dissolved black carbon (DBC) analysis was performed with the benzenepolycarboxylic acid (BPCA) method using the procedures of Dittmar[30] with some modifications[32,40]. One and a half millilitres of methanol eluate was transferred into a 2-ml glass ampoule and dried, and then an additional 1.5 ml of methanol eluate was transferred into the glass ampoule and completely dried. Then, 0.5 ml of concentrated $HNO_3$ was added to the ampoule, and the ampoule was flame-sealed and kept in an oven at 170 °C for 6 h, in accordance with the methods described by Nakane et al.[32]. The ampoule was cooled to room temperature after oxidation; then, $HNO_3$ was evaporated to dryness. The samples in the ampoules were redissolved in 0.2 ml of mobile phase A (as described below) for BPCA analysis using high-performance liquid chromatography (HPLC) with a photodiode array detector (1260 Infinity, Agilent).

The analytical conditions applied for HPLC followed those used by Nakane et al.[32]. Briefly, the samples were eluted at a flow rate of 0.18 ml min$^{-1}$ following gradients from 94% mobile phase A (4 mM tetrabutylammonium bromide, 50 mM sodium acetate and 10% MeOH) to 80% mobile phase B (MeOH). A $C_{18}$ column (3.5 μm, 2.1 × 150 mm, Sunfire, Waters) with a guard column (3.5 μm, 2.1 × 10 mm, Sunfire, Waters) was used for the separation of BPCAs. The injection volume of the sample was 10 μL. The column oven and autosampler of the HPLC system were set to 16 °C and 4 °C, respectively. The absorbance at 235 nm was used to quantify BPCAs.

The DBC concentration was estimated from the BPCA concentrations using Eq. (1) based on Dittmar[30].

$$[DBC] = 12.01 \times 33.4([B6CA] + [B5CA] + 0.5[B4CA] + 0.5[B3CA]) \quad (1)$$

The units of [DBC] and [BPCAs] were μgC L$^{-1}$ and μM, respectively. The ratio of B5CA and B6CA to the total BPCAs ((B5CA + B6CA)/total BPCAs ratio) was calculated as an index of the condensation degree of DBC.

A single seawater sample was used to determine the DBC concentration. The analytical error of the method determined using triplicate seawater samples was <4% in terms of the DBC concentration[32].

**Global removal flux and residence time of dissolved black carbon in the ocean interior**. The global removal flux of DBC (Table 1) was calculated from the basin-scale relationship between the DBC concentration and AOU with the respiration rate in the open ocean[52] as follows:

$$DBC\ removal\ flux = Respiration/12 \times RKR \times SLP \quad (2)$$

where, 12 is the atomic weight of C, RKR is the modified Redfield ratio of $-O_2/C_{org} = 170/117$ (mol-base) in the remineralization processes[57], and SLP is the absolute value of the slope of the linear regression between DBC and AOU with a 95% confidence interval, which is 0.0149–0.0235 (μgC L$^{-1}$/μmol kg$^{-1}$) in the abyssal layer (Fig. 2). For seawater, 1 L of 1 kg was utilised.

The residence time of DBC (Table 1) was estimated by dividing the pool size of DBC by the global removal flux of DBC. The size of the DBC pool was calculated from the size of the DOC pool[43] and a DBC/DOC ratio of 0.02[31].

**Statistical analysis and mapping**. Standard major axis regression and ordinary least squares regression were carried out using the lmodel2 package (version 1.7-3)[58] in R (version 4.0.4). The p value of two-tailed test was reported. Ocean Data View (version 5.1.5)[59] was used to produce the map and the basin-scale distribution of each parameter in Fig. 1 and Supplementary Fig. 1.

**Reporting summary**. Further information on research design is available in the Nature Research Reporting Summary linked to this article.

## Data availability
The data that support the findings of this study are available in the Supplementary Data.

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

## Acknowledgements
We would like to thank the captain, crew and scientists onboard the R/V *Hakuho Maru* for their help with observations, as well as Dr. F. Hashihama for help with water sampling. We are grateful to Drs. H. Obata, M. A. Noguchi, S. Otosaka and T. Hirawake for helpful discussion. This work was supported by the Japan Society for the Promotion of Science grants KAKENHI Nos JP16H02930 (Y.Y.), JP19H04249 (Y.Y.), JP19H05667 (H.O.) and JP19H04260 (H.O.).

## Author contributions
Y.Y. and H.O. contributed to the design of the research. Y.Y., Y.M., J.N. and H.O. contributed to the field observations and sample collections. M.N. and Y.M. performed DBC analyses under Y.Y.'s supervision. Y.Y., M.N. and Y.M. carried out the data analyses. Y.Y. prepared and revised the manuscript with input from J.N. and H.O.

## Competing interests
The authors declare no competing interests.
