## [Peer Review File · Nature Communications]

REVIEWER COMMENTS

Reviewer #1 (Remarks to the Author):

This manuscript presents dissolved black C (DBC) concentrations in a Pacific meridional section from 45S to 55N. The decrease in [DBC] in the abyssal ocean is noted, and compared to AOU, which increases proportionately with latitude northward. This is an impressive data set, which is the largest that I know of for the open ocean. The authors go on to conclude that since AOU is a proxy of the integrated flux of sinking particles, that DBC is removed from the deep ocean to abyssal sediments through adsorption onto sinking particles. They go on to calculate the burial flux of BC to sediments, and report that this is 16-22% of the global BC produced by landscape fire. Though I find the data set good, the interpretation is not as strong as I think it should be to come to this conclusion.

First, I think it is important to provide more statistical tools to the evaluation of the AOU vs [DBC] (Figure 2) and [DBC] vs pre-bomb age of dissolved inorganic carbon (Extended Figure 2). Performing Model II geometric mean regressions of their data would allow calculation of errors of the slope and y-intercepts, which are more conclusive than R² values.

More specifics are needed in the Discussion of water masses. For example, in line 100, they refer to abyssal depths, without providing a depth range for values that in a certain water mass. In the end of the manuscript, they define abyssal depths as >2500m. However, there are three water masses found in this depth range in the Pacific, northward flowing AABW and Circumpolar Deep waters, and southward flowing Pacific deep water. What I think would be better is for the authors to consider defining water masses in terms of density ranges. That way, physical oceanographers will know exactly what water mass they are looking at.

The [DBC] values they report for the deep Pacific (4-6 µgC/L) are lower than those found in a previous study for the eastern North Pacific (15µgC/L, Coppola and Druffel 2016). How do their data compare to those reported by Fang et al. 2017, 2018?

The case for the attribution of the decrease in [DBC] from the South to the North Pacific as having been the result of sorption of DBC onto sinking POC is rather weak. This may be an explanation, but the evidence provided in the manuscript is not convincing that this is the mechanism causing the [DBC] to decrease with latitude north. A correlation between AOU and [DBC] does not prove the causal relationship.

Small comments:

1. Line 203-204: "Three ml of methanol eluate was transferred into a 2-ml glass ampoule.." This is probably a typo.
2. Line 234: It is well known that the spatial distributions of AOU... a reference is needed here.
3. Error bars are needed on the points in Figure 2 and Extended Figure 2, if they are larger than the points.

This paper reports valuable data on the DBC concentrations on a large oceanic scale. It deserves to be published in Nature Geoscience, after major revision.

Reviewer #2 (Remarks to the Author):

Line 12: The statement that BC may control climate appears hyperbolic. It impacts climate.

Line 45: the ocean could only be an ultimate sink for the BC exported to it. Not all BC will get there. Sink should also be defined. Is it a place of final storage or a sink in terms of the global mass balance of BC – i.e. a place of conversion to non-BC (e.g. CO₂).

Line 48-54: Here it is implied that DBC in the ocean is terrestrially derived. This appears not to be the case based upon the stable C isotopes of DBC in the ocean (~-22 to -25) vs rivers (~-28 to -31). See Wagner et al. Nature Communications 2019 (<https://www.nature.com/articles/s41467-019-13111-7>). This paragraph should include this and its ramifications for our understanding of the DBC cycle, including its potential source and fate in the ocean.

General comment on intro so far, a precise, explicit definition of BPCA-derived DBC estimates vs. the other terms in the intro (e.g. BC production on land) needs to be introduced. How comparable are the numbers being compared to DBC? What the author ins contributing to here is a global BPCA budget as part of a global BC budget. Clear definitions, both in terminology and analytical fraction quantified, is required to move the field forward and should be a part of the intro of this paper so readers do not conflate numbers of total BC production on land with BPCA-DBC in the ocean. A recent paper in Nat. Comm. has some comment upon nomenclature etc. (<https://www.nature.com/articles/s41467-021-24477-y>).

Line 105: the statement about dilution not being an explanation for the change in DBC concentration and ratios should either be supported by more data treatment and probably softened to "Thus, dilution is probably not the main factor behind"...

Improved data treatment would include:

Using ^{14}C (and any other available data) as a conservative tracer of dilution rather than apparent age so that DBC and BPCA concs can be plotted directly against the estimated dilution factor. Right now, the Extended Fig 2 plots that show hockey stick plots of DBC conc against ^{14}C age (which is a ratio) are what one would expect for mixing when plotting a concentration vs a quality (e.g. a ratio). A comment upon the potential pitfalls in this type of data treatment is presented in Dittmar et al 2017 (10.1126/science.aam6039).

Instead of saying "ratio of BPCAs" the authors should be explicit (e.g. B_6/B_5 or $\text{B}_6+\text{B}_5/\text{total BPCAs}$).

The authors state that the change in "ratios of BPCAs" is indicative of removal not mixing. This is not a robust statement. Ratios change when you mix two mixtures with different ratios. A more robust way to test this is to show that B_5CA and B_6CA concentrations have different behaviors from one another – i.e. plot each one vs. AOU, ^{14}C and other indicators of water mass mixing or processes.

Line 104: "negatively correlated" – define the relationship – linear, exponential, etc. You would expect the BPCA ratio to vary non-linearly during both mixing and a removal process that fractionates DBC by preferentially removing B_6 . By way of example, see plots of ratios along estuarine mixing gradients (e.g. C/N , d^{13}C) and during photochemical removal (e.g. specifically for $\text{D}_6:\text{B}_5$ see Stubbins et al 2012 and for d^{13}C of DOC Spencer et al 2009).

Line 126: this is too strong a statement about DOC. The suggestion that DBC is removed by sorption is a suggestion, the suggestion that DOC is also removed by this mechanism is a less direct inference from this suggestion. Language should be tempered accordingly.

There is also a circular reasoning to the removal being due to sorption, not microbial or other sinks. The authors say DBC is refractory in the abyssal ocean, a conclusion drawn from its lack of removal in previous data sets of DBC distributions in deep waters. They then reason that when they see removal being correlated with AOU this must be due to sorption, not degradation of DBC, but with no rationale to critically distinguish between sorption and degradation. This needs to be addressed and more nuance introduced into this discussion. This is not as cut and dry as the authors make it sound.

The main piece of evidence that I would use to support the sorption claim is provided by previous research – Coppola et al. (<http://dx.doi.org/10.1002/2013GL059068>). That paper also estimated a global BC burial flux related to DBC sorption to sinking particles and compared this to total ocean C sedimentation and biomass derived BC fluxes globally. Proper citation of this paper and its conclusions would strengthen the author's argument that the AOU-DBC relationship is due to sorption, but also alter the apparent novelty of the paper – given the main finding has been presented before, just via different methods (Coppola et al. used ^{14}C data of sediment BC and sinking particulate organic carbon).

Line 129: Section on Implications of DBC fate in BC cycle

This section does integrate some of the earlier studies that are noted as being overlooked above (e.g. Wagner et al 2019; Coppola et al.). It mentions the Coppola estimate in more detail and it notes the Wagner et al finding that oceanic DBC appears not to be terrigenous. These points would however be best introduced in the introduction as the starting point of knowledge, not produced as potential explanations in the final section.

Final/summary of thoughts:

While the primary finding of the paper, that DBC losses may occur in the deep ocean due to sorption to sinking particles, is not novel, the paper is still sufficiently novel to be of interest to Nature Communications readers. It uses a lot more data and a different approach to reach similar conclusions to Coppola et al. about sorption of DBC. That these two different methods, one based on ^{14}C DBC data for particles and one based on concentrations of DBC, suggest similar fluxes to the ocean floor is one of the strengths of the paper. It suggests that sorption may indeed be a critical sink for DBC in the deep oceans. This similarity in findings and the confirmation of Coppola's estimates by different means should be highlighted more, including by introducing the Coppola study in the introduction both as suggesting a mechanism and as providing an estimate of the flux, both of which the current paper sought to better constrain. This structure to the paper would be easier to follow and a more accurate presentation of the novelty in the current paper.

Most of the comments above can be resolved and should not present a reason to reject the paper. There is plenty that is novel in this paper.

Aron Stubbins

Replies to the comments

Ms. Ref. No.: NCOMMS-21-25492-T

Title: Fate of Dissolved Black Carbon in the Deep Ocean

Authors: Youhei Yamashita, Motohiro Nakane, Yutaro Mori, Jun Nishioka, Hiroshi Ogawa

Replies to the comments of Reviewer #1

This manuscript presents dissolved black C (DBC) concentrations in a Pacific meridional section from 45S to 55N. The decrease in [DBC] in the abyssal ocean is noted, and compared to AOU, which increases proportionately with latitude northward. This is an impressive data set, which is the largest that I know of for the open ocean. The authors go on to conclude that since AOU is a proxy of the integrated flux of sinking particles, that DBC is removed from the deep ocean to abyssal sediments through adsorption onto sinking particles. They go on to calculate the burial flux of BC to sediments, and report that this is 16-22% of the global BC produced by landscape fire. Though I find the data set good, the interpretation is not as strong as I think it should be to come to this conclusion.

Thank you very much for your helpful and constructive review. We have revised the manuscript according to the comments. In particular, we have used Model II geometric mean regression in addition to ordinary least squares regression (R^2) for AOU vs [DBC] etc. As pointed out by the Reviewer #1, we think the analysis make our interpretation more conclusive.

First, I think it is important to provide more statistical tools to the evaluation of the AOU vs [DBC] (Figure 2) and [DBC] vs pre-bomb age of dissolved inorganic carbon (Extended Figure 2). Performing Model II geometric mean regressions of their data would allow calculation of errors of the slope and y-intercepts, which are more conclusive than R^2 values.

Again, thank you very much for this comment. Now, we have added Model II geometric mean (standard major axis) regression for AOU vs [DBC] (revised Fig. 2) and AOU vs BPCAs (revised Fig. 3). In addition to Model II regression, we have added results of ordinary least squares regression (namely R^2 , p) in the figure captions to compare the relationships. We have also revised our estimate about DBC removal flux (revised Table 1 and revised Fig. 4) with change of the regression analysis. We removed analysis regarding with [DBC] vs pre-bomb age of DIC with the revision according to comments from the Reviewer #2.

More specifics are needed in the Discussion of water masses. For example, in line 100, they refer to abyssal depths, without providing a depth range for values that in a certain water mass. In the end of the manuscript, they define abyssal depths as >2500m.

However, there are three water masses found in this depth range in the Pacific, northward flowing AABW and Circumpolar Deep waters, and southward flowing Pacific deep water. What I think would be better is for the authors to consider defining water masses in terms of density ranges. That way, physical oceanographers will know exactly what water mass they are looking at.

Thanks for the comment. We have defined sigma-theta for the intermediate and deep waters and have used these water masses for discussion in the sections entitled “Distribution of DBC in the deep Pacific” and “DBC adsorption onto sinking particles”. According to this revision, we have modified the text (e.g., lines 87-90 in the revised manuscript). We have also added a figure of sigma-theta with schematic pathways of the major deep water masses, namely Circumpolar Deep Water and North Pacific Deep Water (Supplementary Fig. 1a). AABW which is often defined with potential temperature $<0.4^{\circ}\text{C}$ (Ohshima et al., 2013; <https://doi.org/10.1038/ngeo1738>) was not observed during this study. On the other hand, we have used depth based mesopelagic and abyssal layers in the section entitled “Implications of DBC fate in BC cycle”, because the respiration rate (del Giorgio and Duarte, 2002 in the revised Table 1) used to calculate DBC removal flux is depth base. In addition, the description/discussion about abyssal depths as >2500 m was removed because of removal of pre-bomb age of DIC data from the revised manuscript.

The [DBC] values they report for the deep Pacific (4-6 $\mu\text{gC/L}$) are lower than those found in a previous study for the eastern North Pacific (15 $\mu\text{gC/L}$, Coppola and Druffel 2016). How do their data compare to those reported by Fang et al. 2017, 2018?

Coppola and Druffel (2016) determined DBC concentration using a different method than ours. They determined nitrated BPCAs in addition to BPCAs with GC-MS. Since about half of the BPCAs were nitrated (Coppola et al., 2013, <https://doi.org/10.1017/S0033822200048542>), the DBC concentration reported by Coppola and Druffel (2016) is probably higher than our measurements. In addition, the factors estimated DBC concentration from BPCAs were different between Coppola and Druffel (2016) and ours, possibly contributing the difference in DBC concentrations. This difference derived from different BPCA methods are pointed out by a recent study (Fang et al., 2021; <https://dx.doi.org/10.1021/acs.est.0c06386>), suggesting that current estimate of the DBC flux as well as DBC pool size (revised Fig. 4) may be moderate estimate and that DBC probably plays more important role in the global carbon cycle.

The DBC concentrations reported by Fang et al. (2017; 2018) which used the same method with ours were 8.4-10.2 $\mu\text{gC/L}$ for the South China Sea (500–1,500 m) and 5.9 $\mu\text{gC/L}$ for the Circumpolar Deep Water (CDW). Thus, DBC concentrations, in particular, in CDW were comparable between Fang et al (2018) and our study. We have added this point in the revised manuscript (lines 111-113 in the revised manuscript).

The case for the attribution of the decrease in [DBC] from the South to the North Pacific as having been the result of sorption of DBC onto sinking POC is rather weak. This may be an explanation, but the evidence provided in the manuscript is not convincing that this is the mechanism causing the [DBC] to decrease with latitude north. A correlation between AOU and [DBC] does not prove the causal relationship.

We have used Model II regression in the revised manuscript. Therefore, we believe that our conclusions will be more robust. In addition, revised Fig. 3 showed that B5CA+B6CA has a linear relationship with AOU, but B3CA+B4CA does not. Such differences in relationships also support our conclusions. Even that said, we have tempered the conclusion in the revised manuscript. (lines 155-158 in the revised manuscript).

Small comments:

1. Line 203-204: "Three ml of methanol eluate was transferred into a 2-ml glass ampoule.." This is probably a typo.

The sentence has been revised (lines 241-244 in the revised manuscript).

2. Line 234: It is well known that the spatial distributions of AOU... a reference is needed here.

This subsection was deleted accompanying with deletion of ^{14}C of DIC data from the revised manuscript.

3. Error bars are needed on the points in Figure 2 and Extended Figure 2, if they are larger than the points.

Single analysis was carried out to determine the DBC concentration due to limited sample volume. We have added this point in the revised manuscript (line 264 in the revised manuscript). In addition, we have added a result of triplicate analysis using a seawater in the revised manuscript (lines 265-266 in the revised manuscript).

This paper reports valuable data on the DBC concentrations on a large oceanic scale. It deserves to be published in Nature Geoscience, after major revision.

Thank you very much for your thought!

Replies to the comments of Reviewer #2

Line 12: The statement that BC may control climate appears hyperbolic. It impacts climate.

Thanks for the comment. The sentence has been revised accordingly (line 12 in the revised manuscript).

Line 45: the ocean could only be an ultimate sink for the BC exported to it. Not all BC will get there. Sink should also be defined. Is it a place of final storage or a sink in terms of the global mass balance of BC – i.e. a place of conversion to non-BC (e.g. CO₂).

The sentence has been revised accordingly (line 51 in the revised manuscript).

Line 48-54: Here it is implied that DBC in the ocean is terrestrially derived. This appears not to be the case based upon the stable C isotopes of DBC in the ocean (~-22 to -25) vs rivers (~-28 to 31). See Wagner et al. Nature Communications 2019 (<https://www.nature.com/articles/s41467-019-13111-7>). This paragraph should include this and its ramifications for our understanding of the DBC cycle with its potential source and fate in the ocean.

We have introduced Wagner et al (2019) and its ramifications for our understanding of the DBC cycle. We also introduced possible sources of DBC excepting the riverine input based on discussion by Wagner et al (2019)(lines 54-61 in the revised manuscript).

General comment on intro so far, a precise, explicit definition of BPCA-derived DBC estimates vs. the other terms in the intro (e.g. BC production on land) needs to be introduced. How comparable are the numbers being compared to DBC? What the author ins contributing to here is a global BPCA budget as part of a global BC budget. Clear definitions, both in terminology and analytical fraction quantified, is required to move the field forward and should be a part of the intro of this paper so readers do not conflate numbers of total BC production on land with BPCA-DBC in the ocean. A recent paper in Nat. Comm. has some comment upon nomenclature etc. (<https://www.nature.com/articles/s41467-021-24477-y>).

Thank you for the comment. According to the comment, we have clearly mentioned that BC production on land is estimated based on production of char, charcoal, and ash (e.g., lines 27-29 in the revised manuscript). We have also described that a part of char, charcoal, and their leachate is degradable (lines 36-41 in the revised manuscript). Then, we defined DBC and PBC as less reactive, condensed aromatic fractions of BC which can be determined by the benzenepolycarboxylic acid (BPCA) method (lines 42-46 in the revised manuscript). We believe that this revision has made it less misleading to readers. However, unfortunately, we could not evaluate the quantitative linkage between

a global BPCA budget and a global BC budget. Further studies are needed to evaluate the linkage.

Line 105: the statement about dilution not being an explanation for the change in DBC concentration and ratios should either be supported by more data treatment and probably softened to “Thus, dilution is probably not the main factor behind” ...

Improved data treatment would include:

Thanks for the comment. We have realized that it is difficult to evaluate dilution of DBC from ^{14}C of DIC data by this and next comments. Therefore, we have discussed dilution by conservative mixing effect of CDW and NPIW with reference to Hansell and Carlson (2013, <https://doi.org/10.1002/gbc.20067>) in the revised manuscript (lines 128-136 in the revised manuscript). In addition, we have softened the statement about dilution with conservative mixing according to this comment (lines 134-136 in the revised manuscript).

Using ^{14}C (and any other available data) as a conservative tracer of dilution rather than apparent age so that DBC and BPCA concs can be plotted directly against the estimated dilution factor. Right now, the Extended Fig 2 plots that show hockey stick plots of DBC conc against ^{14}C age (which is a ratio) are what one would expect for mixing when plotting a concentration vs a quality (e.g. a ratio). A comment upon the potential pitfalls in this type of data treatment is presented in Dittmar et al 2017 (10.1126/science.aam6039).

Thank you very much for the comment. ^{14}C value of DIC is mainly controlled by radioactive decay and water mass mixing. So, we think that it is not easy to determine the dilution factor from ^{14}C of DIC. Additionally, it is difficult to determine the end-member of DBC concentration for mixing analysis. Therefore, we decided not to use the ^{14}C data in the revised manuscript and deleted original Fig. 1c and Extended Fig. 2 from the revised manuscript. We also deleted description regarding with “Pre-bomb $\Delta^{14}\text{C}$ of dissolved inorganic carbon” and “Removal flux of DBC estimated based on the ^{14}C age of dissolved inorganic carbon” in the Methods section of the revised manuscript. As reply to the former comment, we discussed dilution (conservative mixing) effect by mixing of NPIW and CDW rather than comparison with ^{14}C . It should be noted that our messages/conclusions have not been changed with deletion of ^{14}C data.

Instead of saying “ratio of BPCAs” the authors should be explicit (e.g. B_6/B_5 or $B_6+B_5/\text{total BPCAs}$).

We have clearly mentioned $B_6+B_5/\text{total BPCAs}$ in the text (e.g., lines 118-120 in the revised manuscript) and revised Fig. 1c.

The authors state that the change in “ratios of BPCAs” is indicative of removal not mixing. This is not a robust statement. Ratios change when you mix two mixtures with different ratios. A more robust way to test this is to show that B5CA and B6CA concentrations have different behaviors from one another – i.e. plot each one vs. AOU, I4C and other indicators of water mass mixing or processes.

Thank you very much for the comment. We have plotted B5CA+B6CA vs AOU and B3CA+B4CA vs AOU in the revised Fig. 3. We found that B5CA+B6CA is negatively linearly related to AOU, while B3CA+B4CA is not (Fig. 3 caption). Such differences in the relationship with AOU indicate that highly condensed DBC was preferentially removed from the deep water. We have added this discussion in the revised manuscript (lines 139-142 in the revised manuscript). In addition, we have tempered the discussion/conclusions about the factors that influence the relationship between DBC concentration and AOU in the deep water (lines 143-158 in the revised manuscript).

Line 104: “negatively correlated” – define the relationship – linear, exponential, etc. You would expect the BPCA ratio to vary non-linearly during both mixing and a removal process that fractionates DBC by preferentially removing B6. By way of example, see plots of ratios along estuarine mixing gradients (e.g. C/N, d13C) and during photochemical removal (e.g. specifically for D6:B5 see Stubbins et al 2012 and for d13C of DOC Spencer et al 2009).

According to former comments, we deleted Fig. 2b in the original manuscript and related descriptions from the revised manuscript. Instead, we have added revised Fig. 3 in the revised manuscript.

Line 126: this is too strong a statement about DOC. The suggestion that DBC is removed by sorption is a suggestion, the suggestion that DOC is also removed by this mechanism is a less direct inference from this suggestion. Language should be tempered accordingly.

The descriptions of this paragraph have been tempered according to the comment (lines 159-163 in the revised manuscript).

There is also a circular reasoning to the removal being due to sorption, not microbial or other sinks. The authors say DBC is refractory in the abyssal ocean, a conclusion drawn from its lack of removal in previous data sets of DBC distributions in deep waters. They then reason that when they see removal being correlated with AOU this must be due to sorption, not degradation of DBC, but with no rationale to critically distinguish between sorption and degradation. This needs to be addressed and more nuance introduced into this discussion. This is not as cut and dry as the authors make it sound.

The main piece of evidence that I would use to support the sorption claim is provided by previous research – Coppola et al. (<http://dx.doi.org/10.1002/2013GL059068>). That paper also estimated a global BC burial flux related to DBC sorption to sinking particles and compared this to total ocean C sedimentation and biomass derived BC fluxes globally. Proper citation of this paper and its conclusions would strengthen the author's argument that the AOU-DBC relationship is due to sorption, but also alter the apparent novelty of the paper – given the main finding has been presented before, just via different methods (Coppola et al. used ^{14}C data of sediment BC and sinking particulate organic carbon

Thanks for the comment and suggestion. We have revised this logic thoroughly to remove a circular reasoning. Coppola et al. (2014) was cited properly in the logic to strengthen our arguments. Based on the difference in the relationship between $\text{B5CA}+\text{B6CA}$ vs AOU and $\text{B3CA}+\text{B4CA}$ vs AOU (revised Fig. 3), the possibility that adsorption rather than degradation is the sink for DBC was discussed, citing the references about removal mechanism of PAHs and Coppola et al. (2014). In addition, the conclusion regarding with adsorption onto sinking particles has been tempered (lines 143-158 in the revised manuscript).

Line 129: Section on Implications of DBC fate in BC cycle

This section does integrate some of the earlier studies that are noted as being overlooked above (e.g. Wagner et al 2019; Coppola et al.). It mentions the Coppola estimate in more detail and it notes the Wagner et al finding that oceanic DBC appears not to be terrigenous. These points would however be best introduced in the introduction as the starting point of knowledge, not produced as potential explanations in the final section.

We have introduced Wagner et al (2019) and Coppola et al (2014) in the introduction section (lines 54-61 and 65-67 in the revised manuscript).

Final/summary of thoughts:

While the primary finding of the paper, that DBC losses may occur in the deep ocean due to sorption to sinking particles, is not novel, the paper is still sufficiently novel to be of interest to Nature Communications readers. It uses a lot more data and a different approach to reach similar conclusions to Coppola et al. about sorption of DBC. That these two different methods, one based on ^{14}C DBC data for particles and one based on concentrations of DBC, suggest similar fluxes to the ocean floor is one of the strengths of the paper. It suggests that sorption may indeed be a critical sink for DBC in the deep oceans. This similarity in findings and the confirmation of Coppola's estimates by different means should be highlighted more, including by introducing the Coppola study in the introduction both as suggesting a mechanism and as providing an estimate of the flux, both of which the current paper sought to better constrain. This structure to the

paper would be easier to follow and a more accurate presentation of the novelty in the current paper.

According to the comment and other comments, we have introduced Coppola's study in the Introduction of the revised manuscript (lines 65-67 in the revised manuscript). In addition, in the section entitled "DBC adsorption onto sinking particles", we have mentioned as "In conclusion, the DBC-AOU relationship in the deep ocean (Fig. 2) in combination with the former suggestion by Coppola et al. (2014)²⁷ indicates that adsorption onto sinking particles is probably a major factor decreasing the DBC concentration along with deep-ocean meridional circulation." (lines 155-158 in the revised manuscript).

Most of the comments above can be resolved and should not present a reason to reject the paper. There is plenty that is novel in this paper.

Thank you very much for your review. All comments were good points and very helpful to improve the manuscript.

REVIEWER COMMENTS

Reviewer #3 (Remarks to the Author):

The authors put forth an impressive dataset of DBC along a latitudinal Pacific Ocean transect. In short, they find that DBC concentrations in the abyssal ocean (defined as > 27.5 alpha omega) are significantly negatively correlated with apparent oxygen utilization (AOU). Based upon this regression alone, the authors conclude that DBC is removed in the abyssal ocean via sorption to sinking particles. Well-resolved spatial maps of DBC in the global oceans are sorely needed to grasp black carbon cycling on a global scale and I applaud the authors for their efforts. However, the link between a significant DBC vs. AOU relationship and sorption to sinking particles is still not sufficiently supported and could use further clarification in the text.

Based upon the response to reviewer comments and apparent edits made to the text – The authors have adequately expanded the introduction to summarize and reiterate the necessary background needed to frame DBC cycling and losses on a global scale and have established adequate definitions of water masses studied. However, there are some areas that still lack support, where readers are largely expected to trust that the authors' conclusions are valid, namely that DBC vs. AOU = loss of DBC via sorption to sinking particles. I have listed some areas that need additional improvement below:

Re: terminology – The authors define dissolved and particulate BC as being biorefractory and derived from BPCA analysis. However, the term "BC" is also used to refer to "char, charcoal, and ash" throughout the text, which is confusing to readers and does not do anything to promote consistency within the broader BC literature. For optimal clarity, I strongly recommend that the authors use "pyrogenic carbon" to refer to carbon in bulk pyrolyzed products (char, charcoal, ash) and "black carbon" to refer only to the condensed aromatic fraction of pyrogenic carbon, here specifically quantified and characterized using the BPCA method. Multiple definitions of the same "BC" term in one paper is problematic. As already pointed out by Reviewer 2, refer to a recent discussion on this by Wagner et al. (<https://doi.org/10.1038/s41467-021-24477-y>). For example, L40-41 could read "... and some microbially labile components of pyrogenic carbon, namely dehydrated carbohydrates ..."

L44-45: Need to include a sentence on what the BPCA method is, specifically that it measures BPCAs, which are the oxidation products of condensed aromatic BC structures. Consider that not all readers may understand what the BPCA method is.

L51: "Thus, the ocean can function as a final storage for BC transported to the ocean." – Odd sentence. Delete or revise to clarify.

L56: Change " ^{13}C " to " d^{13}C " and add per mille units to the isotopic values.

L63, 66, and throughout: Consider whether " ^{14}C " and " ^{13}C " can be simply stated as radiocarbon and stable carbon, respectively. These terms are not used often and using fewer abbreviations/shorthand is usually best for a broader, Nat Comm audience, especially since the current study does not incorporate new isotopic data.

L122: Replace "ratio of BPCAs" with " $\text{B6CA}+\text{B5CA}/\text{total BPCAs ratio}$ " – When possible, state directly for optimal clarity and readability. Folks who use the BPCA method report all kinds of "ratios".

L126-136: I see that DOC concentrations are not included in the analysis – Were they measured? It is often useful to contextualize DBC using DOC (e.g., DBC:DOC ratios) to see how the two carbon components covary (or do not covary) along aquatic gradients. If not, it is fine, but including DOC could help support the rejection of dilution and mixing as a possible mechanism for explaining the DBC vs. AOU relationship observed in the current dataset.

L139-142: "An ordinary least squares regression analysis for the deep water showed that $\text{B5CA}+\text{B6CA}$ was negatively linearly related to AOU, while $\text{B3CA}+\text{B4CA}$ was not significantly related to AOU (Fig. 3 caption), indicating that highly condensed DBC was preferentially removed from the deep ocean." Define AOU and how it is being used as a proxy in this specific context, and then how the proxy relates to DBC loss. As written, it is completely unclear to me how a negative DBC vs. AOU relationship clearly indicates DBC removal in the deep ocean – The "in between"

discussion steps are missing.

L149-152: Can support this statement by citing and relating this conclusion to findings by Bostick et al. (2021; <https://doi.org/10.1029/2020JG005981>), who found that BPCAs are relatively biorefractory under controlled experimental conditions.

L179-180: "Assuming that the same proportional relationship between DBC and AOU is applicable to the mesopelagic layer (200–1,000 m) ..." but this proportional relationship does not exist when I plotted up the authors' dataset for mesopelagic samples. To calculate the flux values, should we expect to observe a similarly strong relationship between DBC and AOU between 200 and 1000m? Perhaps I am missing something, but please clarify since it seems like a calculation that can be done with the current set of data, but the authors elect not to include it, which is confusing.

Reviewer 2 stated that "A correlation between AOU and [DBC] does not prove the causal relationship" and the authors replied that they used a slightly different regression model strengthened the argument that DBC is removed via sorption to sinking particles – I don't believe anyone is arguing the overall strength of the apparent relationship between AOU and DBC, but what is still lacking is the supporting detail and connections to justify this main conclusion drawn. As was originally mentioned by Reviewer 2, the main takeaway from this paper is that the authors arrived at the same flux value as Coppola et al. using a new and more efficient quantitative approach. The findings themselves are not novel, except for the significant expansion of DBC spatial data in the Pacific, but they do strengthen constraints on possible removal mechanisms (sorption to particles) for DBC in the deep ocean. It is also informative to describe alternative mechanisms that are possible and cannot be ruled out with one regression analysis (e.g., biodegradation) – This opens up avenues of future research too.

Replies to the comments

Ms. Ref. No.: NCOMMS-21-25492-A

Title: Fate of Dissolved Black Carbon in the Deep Ocean

Authors: Youhei Yamashita, Motohiro Nakane, Yutaro Mori, Jun Nishioka, Hiroshi Ogawa

Replies to the comments of Reviewer #3

The authors put forth an impressive dataset of DBC along a latitudinal Pacific Ocean transect. In short, they find that DBC concentrations in the abyssal ocean (defined as > 27.5 alpha omega) are significantly negatively correlated with apparent oxygen utilization (AOU). Based upon this regression alone, the authors conclude that DBC is removed in the abyssal ocean via sorption to sinking particles. Well-resolved spatial maps of DBC in the global oceans are sorely needed to grasp black carbon cycling on a global scale and I applaud the authors for their efforts. However, the link between a significant DBC vs. AOU relationship and sorption to sinking particles is still not sufficiently supported and could use further clarification in the text.

Based upon the response to reviewer comments and apparent edits made to the text – The authors have adequately expanded the introduction to summarize and reiterate the necessary background needed to frame DBC cycling and losses on a global scale and have established adequate definitions of water masses studied. However, there are some areas that still lack support, where readers are largely expected to trust that the authors' conclusions are valid, namely that DBC vs. AOU = loss of DBC via sorption to sinking particles. I have listed some areas that need additional improvement below:

Thank you very much for your helpful and constructive review. We have revised the manuscript according to the comments. In particular, we have thoroughly rewritten the discussion about the link between a significant DBC vs. AOU relationship and DBC sorption onto sinking particles to make our message clearer. Please see details below.

We have replaced “adsorption” with “sorption” in the revised manuscript based on reviewers' comments and Coppola et al. (2014).

Re: terminology – The authors define dissolved and particulate BC as being biorefractory and derived from BPCA analysis. However, the term “BC” is also used to refer to “char, charcoal, and ash” throughout the text, which is confusing to readers and does not do anything to promote consistency within the broader BC literature. For optimal clarity, I strongly recommend that the authors use “pyrogenic carbon” to refer to carbon in bulk pyrolyzed products (char, charcoal, ash) and “black carbon” to refer only to the condensed aromatic fraction of pyrogenic carbon, here specifically quantified and characterized using the BPCA method. Multiple definitions of the same “BC” term in one paper is problematic. As already pointed out by Reviewer 2, refer to a recent discussion on this by Wagner et al.

(<https://doi.org/10.1038/s41467-021-24477-y>). For example, L40-41 could read “... and some microbially labile components of pyrogenic carbon, namely dehydrated carbohydrates ...”

Thank you very much for the comment. According to the comment, we have replaced BC with pyrogenic carbon (PyC) for the bulk pyrolyzed products (char, charcoal, ash) in the revised manuscript (e.g., lines 25-41 in the revised manuscript).

L44-45: Need to include a sentence on what the BPCA method is, specifically that it measures BPCAs, which are the oxidation products of condensed aromatic BC structures. Consider that not all readers may understand what the BPCA method is.

A description, what the BPCA method is, has been added in the revised manuscript (lines 43-44 in the revised manuscript).

L51: “Thus, the ocean can function as a final storage for BC transported to the ocean.” – Odd sentence. Delete or revise to clarify.

The sentence has been deleted from the revised manuscript.

L56: Change “13C” to “d13C” and add per mille units to the isotopic values.

We have replaced 13C with stable carbon throughout the manuscript according to the next comment. We have added ‰ to the isotopic values in the revised manuscript (line 55-56 in the revised manuscript).

L63, 66, and throughout: Consider whether “14C” and “13C” can be simply stated as radiocarbon and stable carbon, respectively. These terms are not used often and using fewer abbreviations/shorthand is usually best for a broader, Nat Comm audience, especially since the current study does not incorporate new isotopic data.

We have replaced 14C and 13C with radiocarbon and stable carbon, respectively, throughout the revised manuscript (e.g., line 55 in the revised manuscript).

L122: Replace “ratio of BPCAs” with “ B6CA+B5CA/total BPCAs ratio” – When possible, state directly for optimal clarity and readability. Folks who use the BPCA method report all kinds of “ratios”.

The sentence has been amended (line 123 in the revised manuscript). The similar descriptions have also been amended throughout the revised manuscript.

L126-136: I see that DOC concentrations are not included in the analysis – Were they measured? It is often useful to contextualize DBC using DOC (e.g., DBC:DOC ratios) to see how the two carbon components covary (or do not covary) along aquatic gradients. If not, it is fine, but including DOC could help support the rejection of

dilution and mixing as a possible mechanism for explaining the DBC vs. AOU relationship observed in the current dataset.

Thanks for the comment. Yes, DOC data is useful for the analysis. However, unfortunately, we don't have DOC concentration data for samples reported in the current manuscript.

L139-142: “An ordinary least squares regression analysis for the deep water showed that B5CA+B6CA was negatively linearly related to AOU, while B3CA+B4CA was not significantly related to AOU (Fig. 3 caption), indicating that highly condensed DBC was preferentially removed from the deep ocean.” Define AOU and how it is being used as a proxy in this specific context, and then how the proxy relates to DBC loss. As written, it is completely unclear to me how a negative DBC vs. AOU relationship clearly indicates DBC removal in the deep ocean – The “in between” discussion steps are missing.

Thank you very much for the comment. According to the comments, we have re-organized this section to make our conclusion clearer.

About this sentence, the AOU was defined here as a function of oxygen utilization rate and time after the formation of deep water (lines 148-149 in the revised manuscript). Since it has been suggested that low molecular weight polycyclic aromatic hydrocarbons are preferentially degraded by microbes (ref. 45, 46), it is expected that B3CA+B4CA, which is preferentially derived from low molecular weight DBC, is negatively related to AOU (a proxy for time) in the deep ocean when DBC is subjected to microbial degradation. However, the B3CA+B4CA is not related to AOU in the deep water (Fig. 3a), indicating that microbial degradation is probably not a major factor controlling the DBC concentration in the deep ocean. We have added this discussion in the revised manuscript (lines 143-151 in the revised manuscript).

L149-152: Can support this statement by citing and relating this conclusion to findings by Bostick et al. (2021; <https://doi.org/10.1029/2020JG005981>), who found that BPCAs are relatively biorefractory under controlled experimental conditions.

Thanks for the suggestion. We have cited the paper in the revised manuscript (lines 151-153 in the revised manuscript).

L179-180: “Assuming that the same proportional relationship between DBC and AOU is applicable to the mesopelagic layer (200–1,000 m) ...” but this proportional relationship does not exist when I plotted up the authors’ dataset for mesopelagic samples. To calculate the flux values, should we expect to observe a similarly strong relationship between DBC and AOU between 200 and 1000m? Perhaps I am missing something, but please clarify since it seems like a calculation that can be done with the current set of data, but the authors elect not to include it, which is confusing.

Thank you very much for confirming the DBC and AOU relationship in the mesopelagic layer. The DBC and AOU relationship is evident in the abyssal layer because the influence of water mass mixing is relatively small in the abyssal layer. In the mesopelagic layer, the negative relationship between DBC and AOU may be masked by complex mixing of intermediate waters (AAIW and NPIW) with other water masses (deep water and mode waters), all of which have different levels of surface water-derived DBC. However, removal of DBC through sorption onto sinking particles most likely occurs not only the abyssal layer but also the mesopelagic layer. Furthermore, the relationship between DBC and AOU may be similar between the abyssal and mesopelagic layers if we consider that the removal of DBC and the increase in AOU are the result of sorption to fast-sinking particles and the decomposition of slow-sinking particles, respectively (please see our reply to the next comment). Therefore, we could assume that the same proportional relationship between DBC and AOU is applicable to the mesopelagic layer. We have added this discussion in the revised manuscript (lines 192-198 in the revised manuscript).

Reviewer 2 stated that “A correlation between AOU and [DBC] does not prove the causal relationship” and the authors replied that they used a slightly different regression model strengthened the argument that DBC is removed via sorption to sinking particles – I don’t believe anyone is arguing the overall strength of the apparent relationship between AOU and DBC, but what is still lacking is the supporting detail and connections to justify this main conclusion drawn. As was originally mentioned by Reviewer 2, the main takeaway from this paper is that the authors arrived at the same flux value as Coppola et al. using a new and more efficient quantitative approach. The findings themselves are not novel, except for the significant expansion of DBC spatial data in the Pacific, but they do strengthen constraints on possible removal mechanisms (sorption to particles) for DBC in the deep ocean. It is also informative to describe alternative mechanisms that are possible and cannot be ruled out with one regression analysis (e.g., biodegradation) – This opens up avenues of future research too.

Thank you very much for the comment. AOU represents the total amount of oxygen consumed after the formation of deep water, because AOU is function of oxygen utilization rate and time after the formation of deep water. In the deep ocean, AOU mainly increases through the microbial degradation of sinking particulate organic carbon (POC) rather than of DOC (ref. 48). AOU is also likely related to the integrated volume of sinking particles that pass through the water mass, because not all sinking particles degrade in the water mass. However, actual mechanism for the relationships between DBC and AOU is more complicated, because a substantial fraction of the sinking particles that pass the water mass degrade before reaching the sediment and likely release DBC if DBC is sorbed. The actual mechanism is probably explained by a two component model of sinking particle, i.e., fast-sinking and slow-sinking particles

(ref. 49, 50). AOU increases mainly with degradation of slow-sinking particles, while DBC is mostly removed through sorption onto fast-sinking particles that settle without undergoing extensive degradation. The affinity of the fast-sinking particles for hydrophobic DOC including DBC attenuate with increasing the water depth due to their sorption in accordance with first-order reaction. This attenuation pattern is probably similar to exponential decrease in sinking POC flux with increasing the water depth (ref. 49). These similar attenuation patterns contribute the apparent relationship between AOU and DBC. Thus, the removal of the DBC from the water column through sorption onto sinking particles is probably a major factor contributing the negative relationship between DBC and AOU in the deep water (Fig. 2). This conclusion is supported by the negative relationship between B5CA+B6CA and AOU (Fig. 3b) as suggested by Coppola et al. (2014)(ref. 27). We have written this discussion thoroughly in the revised manuscript (lines 154-171 in the revised manuscript). We believe that our conclusion/discussion has been made clearer by this revision.

According to the comment regarding with alternative mechanisms that are possible and cannot be ruled out with one regression analysis, we have added descriptions in the revised manuscript (line 176-179 in the revised manuscript).

REVIEWERS' COMMENTS

Reviewer #3 (Remarks to the Author):

The authors have sufficiently addressed my concerns and I have no major comments on the current draft. Minor grammatical comments are below:

L164 - Missing word? "... that pass THROUGH the water mass ..."

L196 - "mode waters" - I've never heard this phrase before, is it correct? I also recognize I may not be aware of all oceanographic terminology.

L210 - Revise to "... the DBC APPARENT RADIOCARBON ages of >20,000 ..."

Replies to the comments

Ms. Ref. No.: NCOMMS-21-25492-B

Title: Fate of dissolved black carbon in the deep Pacific Ocean

Authors: Youhei Yamashita, Motohiro Nakane, Yutaro Mori, Jun Nishioka, Hiroshi Ogawa

Replies to the comments of Reviewer #3

The authors have sufficiently addressed my concerns and I have no major comments on the current draft. Minor grammatical comments are below:

Thank you very much for your careful reviews. We have revised the manuscript according to the comment.

L164 - Missing word? "... that pass THROUGH the water mass ..."

Amended accordingly (line 168 in the revised manuscript).

L196 - "mode waters" - I've never heard this phrase before, is it correct? I also recognize I may not be aware of all oceanographic terminology.

"mode waters" has been replaced with "overlying water masses" (line 200 in the revised manuscript).

L210 - Revise to "... the DBC APPARENT RADIOCARBON ages of >20,000 ..."

Amended accordingly (line 214 in the revised manuscript). We have also added "apparent" to other descriptions about radiocarbon age (lines 63 and 67 in the revised manuscript).